# Magnetic Polyion Complex Micelles for Cell Toxicity Induced by Radiofrequency Magnetic Field Hyperthermia

**DOI:** 10.3390/nano8121014

**Published:** 2018-12-06

**Authors:** Vo Thu An Nguyen, Marie-Claire De Pauw-Gillet, Mario Gauthier, Olivier Sandre

**Affiliations:** 1University Bordeaux, LCPO, UMR 5629, F-33600 Pessac, France; annguyenvothu@gmail.com; 2Department of Chemistry, University of Waterloo, Waterloo, ON N2L 3G1, Canada; 3CNRS, Laboratoire de Chimie des Polymères Organiques, UMR 5629, F-33600 Pessac, France; 4University of Liège, Mammalian Cell Culture Laboratory, GIGA-R, B-4000 Liège, Belgium; Marie-Claire.depauw@ulg.ac.be

**Keywords:** magnetic polyion complex micelle, arborescent polymer, double-hydrophilic block copolymer, cytotoxicity, cell internalization, magnetic field hyperthermia

## Abstract

Magnetic nanoparticles (MNPs) of magnetite (Fe_3_O_4_) were prepared using a polystyrene-*graft*-poly(2-vinylpyridine) copolymer (denoted G0PS-*g*-P2VP or G1) as template. These MNPs were subjected to self-assembly with a poly(acrylic acid)-*block*-poly(2-hydroxyethyl acrylate) double-hydrophilic block copolymer (DHBC), PAA-*b*-PHEA, to form water-dispersible magnetic polyion complex (MPIC) micelles. Large Fe_3_O_4_ crystallites were visualized by transmission electron microscopy (TEM) and magnetic suspensions of MPIC micelles exhibited improved colloidal stability in aqueous environments over a wide pH and ionic strength range. Biological cells incubated for 48 h with MPIC micelles at the highest concentration (1250 µg of Fe_3_O_4_ per mL) had a cell viability of 91%, as compared with 51% when incubated with bare (unprotected) MNPs. Cell internalization, visualized by confocal laser scanning microscopy (CLSM) and TEM, exhibited strong dependence on the MPIC micelle concentration and incubation time, as also evidenced by fluorescence-activated cell sorting (FACS). The usefulness of MPIC micelles for cellular radiofrequency magnetic field hyperthermia (MFH) was also confirmed, as the MPIC micelles showed a dual dose-dependent effect (concentration and duration of magnetic field exposure) on the viability of L929 mouse fibroblasts and U87 human glioblastoma epithelial cells.

## 1. Introduction

Superparamagnetic iron oxide nanoparticles (SPIONs) have received significant attention for uses in targeted drug delivery [1,2,3], as magnetic resonance imaging contrast agents [4,5] and as heat mediators in the magnetic hyperthermia treatment of tumours [6,7,8]. Among numerous chemical techniques by which SPIONs can be synthesized, the ferric/ferrous ions alkaline coprecipitation method has been preferred as the simplest and most efficient route for large scale production [8]. Hydrophilic SPIONs (with rather broad size distributions and an average diameter of 7–8 nm, as measured by X-ray diffraction) are generally produced by the alkaline coprecipitation of FeCl_3_ and FeCl_2_ salts, as first reported by Massart [9]. The main disadvantage of alkaline coprecipitation is the lack of control over the particle size distribution, because nucleation and crystal growth occur simultaneously. Decoupling of the nucleation process from crystal growth is mandatory for the formation of iron oxide nanoparticles of uniform size [10]. Control over the size and the morphology of MNPs may be achieved by tuning the nucleation process through adjustment of the geometry and size of organic templates with molecular structures strongly interacting with the iron oxide precursors introduced in the first step of MNP synthesis [11]. Well-defined polymer structures have a stronger impact on the outcome of size- and shape-controlled MNPs than simple homopolymer chains [11]. In this study, we used a G1 arborescent copolymer, G0PS-*g*-P2VP, consisting in a G0 or comb-branched polystyrene substrate grafted with 182 P2VP side-chains and having a 2VP unit content of 91 mol%, synthesized by Gauthier et al. [12], as a template for the Fe_3_O_4_ NP synthesis. The high content of the copolymers in 2VP, a ligand that coordinates iron cations, combined with the very compact structure of the G1 arborescent copolymer, should make it a very good platform for the complexation of iron (ferrous and ferric) cations and templating of the MNPs. The resulting MNPs were studied by dynamic light scatting (DLS) and thermogravimetric analysis (TGA), while the crystallites, most probably a mixture of magnetite (Fe_3_O_4_) and maghemite (γ-Fe_2_O_3_), were evidenced by selected area electron diffraction (SAED) and their size measured by TEM to demonstrate the templating effect of the copolymers used.

Under physiological conditions (pH ~ 7, 150 mM ionic strength), uncoated iron oxide MNPs precipitate under the influence of van der Waals forces, or when subjected to a magnetic field inducing magnetic dipolar attraction, due to their insufficient electrostatic repulsive interactions that are screened at high ionic strength. In a clinical MRI device, for example, precipitation and agglomeration of the iron oxide MNPs can lead to clotting in blood vessels [13] or increased cytotoxicity [14]. The stabilization of magnetic suspensions is thus needed and can be achieved most conveniently by introducing electrostatic and/or steric repulsions between the MNPs. The well-documented sensitivity of electrostatic forces to the ionic strength of the dispersion environment leaves steric stabilization as the preferred option to enhance the colloidal stability of MNP suspensions. The chemical nature of the coating on MNPs has a strong, if not vital impact on their cellular uptake efficiency, cytotoxicity, biocompatibility and biodegradability once they are exposed to physiological conditions [15]. Fe_3_O_4_ MNPs coated with poly(ethylene glycol) (PEG) chains [16], for example, displayed no apparent cytotoxicity in in vitro assays, owing to the biocompatible coating preventing the iron oxide core from interacting with the cells. Silica-coated iron oxide NPs, Ferumoxsil^®^ (AMI-121), has also been tested in clinical trials by oral administration [17] and approved against bowel cancer [18]. Commercially available contrast agents such as Ferridex^®^, Resovist^®^, Supravist^®^ and Sinerem^®^ are coated with dextran or carboxy-dextran polysaccharides [19,20], which are unfortunately prone to detachment and aggregation due to weak binding of the coating materials to iron oxide MNPs (simple adsorption). Dextran-magnetite MNPs caused cell death and reduced cell proliferation as strongly as uncoated iron oxide particles [21], because the dextran coating can collapse and expose iron oxide MNP aggregates to the cellular components, leading to membrane disruption presumably through lipid peroxidation by reactive oxygen species (ROS) produced by the Fe^2+^ ions (Fenton reaction) [22,23]. Another polysaccharide reported in the literature for MNP coating is chitosan, which was proved to induce colloidal stability, high cell internalization and no cytotoxicity [24]. As reviewed by Soenen and De Cuyper, the adverse effects of iron oxide MNPs must indeed be carefully studied through in vitro toxicity assays before administration in vivo and in particular their intracellular concentration and biodegradation deserve much attention [25].

Considering the composition of the arborescent copolymer G0PS-*g*-P2VP (or in short G1) used as template for MNP synthesis, whose 2VP units can be protonated, a polymer stabilizing shell bound through polyelectrolyte complexation already reported for copolymer micelles [26] appears promising, as it offers stronger interactions rather than mere physical adsorption and is much more versatile than covalent bonding. In this article, the polyion complexation process of G1 unimolecular micelles with PAA-*b*-PHEA DHBC chains was adapted from our previous study [27] to produce MPIC micelles, noted G1@Fe_3_O_4_, containing MNPs synthesized in situ by a templated coprecipitation route. The colloidal stability of MPIC micelles at pH 7 was studied by dynamic light scattering (DLS), while their morphology was observed by atomic force microscopy (AFM) imaging.

The cytotoxicity and cell internalization of the MPIC micelles were evaluated in vitro to predict toxicological consequences of the MNPs in vivo and to reveal the fate of the MNPs once they enter the organism. An in vitro study of magnetic field hyperthermia (MFH) was also performed on two different cell lines selected for different reasons: mouse L929 fibroblasts, a non-cancerous immortalized cell line recommended in the ISO10993-1: 2009 procedure “Biological evaluation of medical devices” [28] and in view of the foreseen medical application, U87 human glioblastoma epithelial cells, the high grade brain tumour cells targeted in the first clinical trials with magnetic hyperthermia used in combination with conventional radiotherapy [29]. In both cases, a colloidal suspension of MNPs (the MPIC micelles) was incubated with the cells, allowing the MNPs to be internalized through the cell membrane. The application of an external alternating magnetic field (AMF) induced heat in the MNPs, intended to kill the cells. In our work, the incubation concentration of MPIC micelles and the AMF exposure time strongly influenced the viability of L929 and U87 cells. Unlike the pioneering work of Fortin et al. [30] on AMF-induced hyperthermia performed on cell pellets (leading to a high iron oxide concentration of up to 1.5 mg·mL^−1^, compatible with a macroscopic temperature rise), our study was performed on cell suspensions, at lower MNP concentrations not leading to macroscopic heating of the solvent. Cell AMF-induced cell toxicity yet without increase of temperature, ascribed to thermal losses in the cell aqueous medium of high thermal conductivity, is discussed at the end of the article within the context of recent published works on intracellular magnetic hyperthermia.

## 2. Materials and Methods

### 2.1. Materials

The materials used are listed in the Appendix A.

### 2.2. Synthesis of MPIC Micelles

**Synthesis of polymers.** The G1 or G0PS-*g*-P2VP copolymer (${\bar{M}}_{n}$ = 1.1 × 10^6^ g·mol^−1^; ${\bar{M}}_{w}$/${\bar{M}}_{n}$ = 1.08) was synthesized by Gauthier et al. [12] by anionic polymerization and grafting. The preparation of double-hydrophilic block copolymers (DHBC) PAA_13_-*b*-PHEA_150_ (${\bar{M}}_{w}$/${\bar{M}}_{n}$ = 1.23), PAA_27_-*b*-PHEA_260_ (${\bar{M}}_{w}$/${\bar{M}}_{n}$ = 1.16) and the fluorescently labelled derivative PAA_27_*-*b*-PHEA_260_ was achieved by atom-transfer radical polymerization (ATRP) of the corresponding protected monomers (*tert*-butyl acrylate and 2-trimethylsilyloxyethyl acrylate, respectively) as described previously [27] and carbodiimide hydrochloride (EDC) activated amide bond formation with fluoresceinamine (Appendix A).

**Synthesis of magnetic nanoparticles (MNPs) using G1 as template.** A stock solution of FeCl_3_ (24 mg, 0.148 mmol) and FeSO_4_·7H_2_O (24.2 mg, 0.087 mmol) in 2 mL of Milli-Q water acidified with HCl (pH 1.4) was prepared freshly before use. In a 20 × 150 mm^2^ test tube, G1 (35 mg, 0.333 mmol of 2VP units) was completely dissolved in 6 mL of aqueous HCl solution at pH 1.4 by sonication (37 kHz, 30 min, Ultrasonic Cleaner Elmasonic SH075EL). The G1 solution was then stirred vigorously with a mechanical rotor (>800 rpm) in an oil bath at either 50 or 80 °C and 2 mL of the ferrous and ferric mixed salt stock solution was quickly added (total Fe/2VP molar ratio = 0.7), followed by 170 µL of concentrated NH_4_OH to trigger the coprecipitation into colloidal Fe_3_O_4_. The solution colour quickly changed from yellow to black. Sedimentation of the MNPs in the magnetic field gradient of a strong NdFeB permanent magnet allowed complete removal of the excess reagents. The MNPs were rinsed with water (2 × 15 mL), acetone (2 × 15 mL) and diethyl ether (2 × 15 mL) before redispersion in 20 mL of dilute HNO_3_ (pH 1.4), to produce a black homogenous ferrofluid that was a relatively stable colloidal dispersion of MNPs (as long as not submitted to a magnetic field gradient).

**Synthesis of magnetic polyion complex (MPIC) micelles.** The procedure for the complexation of the G1-templated MNPs G1@Fe_3_O_4_ with PAA-*b*-PHEA block copolymers to obtain MPIC micelles in aqueous solutions was adapted from our previous study [27]. The parameter *f* = CO_2_H/N was used to quantify the molar ratio of CO_2_H groups relatively to 2VP units. The following procedure describes the complexation of the G1-templated MNPs G1@Fe_3_O_4_ with PAA_27_-*b*-PHEA_260_ for *f* = 0.5. In a 10-mL vial, G1@Fe_3_O_4_ (0.91 mg Fe_3_O_4_, 1.75 mg of G1, 16.7 µmol of N functional groups) was readily dispersed in 1 mL of HNO_3_ solution (pH 1.4). A pH 7 aqueous solution of PAA_27_-*b*-PHEA_260_ (9.5 mg, 5 mg/mL, 8.3 µmol of CO_2_H functional groups, *f* = CO_2_H/N = 0.5) was quickly added, the mixture was stirred on a magnetic stirrer for 1 h before the pH was adjusted to 4.7 with a 1 M NaOH solution and stirring was continued for 1 h. The pH was further adjusted to 7 with 0.1 M NaOH and the solution was stirred for 30 min. The solution was then dialyzed (50,000 MWCO Spectra/Por^®^ 7 regenerated cellulose bag) against Milli-Q water (5 L) for 24 h before it was collected and stored at 4 °C. The preparation of the fluorescently labelled MPIC micelles (hereinafter referred to as MPIC* micelles) was performed in the dark by the same procedure but with a mixture of 5% PAA_27_^*^-*b*-PHEA_260_ (7.0% labelled) and 95 mol% of (non-labelled) PAA_27_-*b*-PHEA_260_.

### 2.3. Characterization of MPIC Micelles

MPIC micelles were characterized using Dynamic Light Scattering (DLS), Atomic Force Microscopy (AFM), Transmission Electron Microscopy (TEM), Selected Area Electron Diffraction (SAED) and Thermogravimetric Analysis (TGA) (Appendix A).

### 2.4. Biocompatibility Assessment

The mouse fibroblast-like cell line L929 was purchased from the European Collection of Cell Cultures (85011425, ECACC, Public Health England, Salisbury, UK). The U87 human glioblastoma was the line *ATCC*^®^ HTB-14™ obtained from the American Tissue Culture Collection (Manassas, VA, USA). The materials, cell culture, cytotoxicity and cell internalization studies using confocal laser scanning microscopy, TEM and Fluorescence-Activated Cell Sorting (FACS) of MPIC micelles in cells are described in Appendix A.

### 2.5. In Vitro Magnetic Field Hyperthermia

The L929 cells and U87 cells were seeded in a 48-well plate at a density of 2 × 10^4^ cells/0.5 mL/well and allowed to grow at 37 °C in a 5% CO_2_ humidified incubator. The MPIC micelles were prepared in DMEM complete medium at 1250, 700 and 140 µg of Fe_3_O_4_/mL before cell treatment. After 6 h the DMEM complete medium was replaced with 0.5 mL of MPIC micelle suspension and the control cells were incubated in complete medium alone. After 15 h of exposure to the MNPs, the suspension was removed and the cells were rinsed twice with PBS (Ca^2+^/Mg^2+^ free) solution to remove non-uptaken MNPs. After being detached with trypsin 10× solution diluted 10-fold in PBS 1 × (Ca^2+^/Mg^2+^ free) solution, both controls and treated cells were centrifuged at 200 rcf for 5 min. To minimize the effects of magnetic field inhomogeneity a small sample volume (0.35–0.4 mL) is recommended [30], so the cells were suspended in small NMR plastic tubes. This choice was also made to control the local magnetic field lines experienced by the cells, through well-defined sample geometry (cylindrical shape with a large aspect ratio), location in the solenoid (6 NMR 0.4 mL tubes can be placed simultaneously near the centre of the coil) and orientation (parallel to the coil axis). This precaution minimizes the so-called “demagnetization effect” lowering the local magnetic field as compared to the applied field strength. On the other hand, the diameter (3 mm) was not so small as to increase the effect of thermal losses. Each magnetic field treatment was applied on 6 tubes (3 samples, 3 controls). The trypsin solution was removed before the cells were redispersed in 0.35 mL of DMEM complete medium (containing 10 mM HEPES buffer, to maintain the pH in the absence of a controlled CO_2_ pressure) and transferred to the tubes previously sterilized by UV exposure. These tubes, containing cells with or without internalized MPIC micelles (controls), were placed in a small bath of deionized water inside a double-walled glass jacket maintained at a temperature of 37 °C during the treatment. The whole system was inserted in an induction copper coil (4 turns with 10 mm inter-loop spacing and 55 mm diameter, cooled by internal water circulation at 18 °C, as shown in Appendix A).

The coil was driven by a current of 234 A at full power of the generator (3 kW MOSFET solid state resonant circuit, Seit Elettronica Junior™, Treviso, Italy) to produce a high frequency alternating magnetic field (AMF) at frequency *f* = 755 kHz and root mean square (RMS) magnetic field strength *H*_max_ = 10.2 kA·m^−1^ according to numerical simulations by a finite element method [31]. Two RF magnetic field exposure times were tested: 1.5 and 3 h. The temperature of the media in one control sample and in one treated sample was monitored simultaneously with two fibre optic thermometers (OTG-M420, Opsens, Québec, QC, Canada). After exposure to the RF magnetic field, the cells were reseeded in a 48-well plate and incubated at 37 °C in a 5% CO_2_ humidified incubator. After 15 h of incubation, the complete medium was removed and the cells were rinsed with PBS (with Ca^2+^/Mg^2+^) solution (0.5 mL/well) before the addition of 60 μL of MTS solution (prior dilution in 300 μL of PBS with Ca^2+^/Mg^2+^). The plates were incubated at 37 °C for 120 min before measurement of the absorbance at 490 nm. This MTS assay was preferred to the more classical MTT test for cell viability, as it avoids the step of formazan dissolution into DMSO [32]. The results were expressed as the percentage of intracellular reductase activity for the treated cells relatively to the untreated cells (control, 100% viability). Independent experiments were performed twice (*n* = 2).

## 3. Results and Discussion

### 3.1. Templated In Situ Coprecipitation

Due to its high content (91 mol%) of 2VP units (a strong iron-coordinating ligand) grafted in a very compact structure with a uniform size (*D*_h_ = 29 ± 4 nm, PDI = 0.04, Figure 1a; *D*_TEM_ = 22 ± 3 nm, Figure 2a), the G1 arborescent polymer should be a useful platform for templating iron oxide NP formation. After redispersion in HNO_3_ solution (pH 1.4), the MNPs prepared by in situ alkaline coprecipitation of ferrous and ferric salts within the template, G1@Fe_3_O_4_, had better colloidal stability than bare Fe_3_O_4_ MNPs prepared as a control sample under the same conditions but without G1 arborescent copolymer. The bare Fe_3_O_4_ MNPs clustered and aggregated after a short time period (10 min), or immediately when subjected to a weak magnetic field, while the G1@Fe_3_O_4_ suspension was stable in both cases (Appendix A).

The hydrodynamic diameter of the bare Fe_3_O_4_ MNPs (*D*_h_ ≈ 1 µm, PDI = 0.8) determined by DLS analysis in HNO_3_ at pH 1.4 was much larger and much less uniform than for G1@Fe_3_O_4_, with *D*_h_ = 131 nm, PDI = 0.286 (Figure 1a). In an acidic environment, the Fe-OH moieties at the surface of the iron oxide MNPs become protonated into Fe-OH_2_^+^, inducing electrostatic repulsions between the particles. However these electrostatic repulsions were insufficient to stabilize the system, resulting in clustering and aggregation. For G1@Fe_3_O_4_, the much better colloidal stability achieved is attributed to the additional repulsive forces originating from the protonated 2VP units, together with steric repulsion. The G1 copolymer acted as a template encapsulating the Fe_3_O_4_ precursors and stabilizing the MNPs to some extent after their in situ synthesis. The discrepancy between the intensity-weighted *D*_h_ = 131 nm and number-weighted $D_{h}^{n}$ = 92 nm, combined with a broad size distribution (PDI > 0.2), nonetheless indicated some level of aggregation.

To emphasize the advantages of the G1 arborescent copolymer as a template, alkaline coprecipitation in the presence of linear poly(4-vinylpyridine) (P4VP, ${\bar{M}}_{n}$ = 32,000 g·mol^−1^, ${\bar{M}}_{w}$ = 65,000 g·mol^−1^) was also carried out for comparison and TEM was used to determine the crystallite size of the Fe_3_O_4_ NPs (composed of both crystalline and amorphous components). The size measured by TEM is a number-weighted value, thus the size analysis was performed on a large number of particles (≥100) to obtain meaningful statistical results. The crystallite size of P4VP@Fe_3_O_4_ determined by TEM analysis was 7.0 ± 1.4 nm (Figure 2b), which is typical for samples prepared by non-templated alkaline coprecipitation [9]. The crystallite size of the Fe_3_O_4_ NPs increased to 9.1 ± 1.7 nm in the presence of the G1 template at 50 °C (Figure 2c). The encapsulation of Fe^2+/3+^ ions within the G1 micelle template presumably distributed the ions into a smaller volume, which allowed a short burst in the nucleation rate as compared to homogeneous nucleation. In comparison to homogenous nucleation, heterogeneous nucleation within the P2VP domain of the G1 micelles forces the Fe^2+^ and Fe^3+^ cations closer to each other, since complexation partly overcomes their electrostatic repulsion. In doing so, the nucleation and growth steps are more likely to be separated.

Growth of the Fe_3_O_4_ crystallites was further increased to 12.1 ± 2.0 nm when G1-templated coprecipitation was performed at 80 °C (Figure 2d). The elevated temperature probably accelerated both the nucleation and growth steps, resulting in larger crystallites. An attempt at 100 °C led to a higher oxidation level: the suspension had a brownish colour typical of the maghemite phase γ-Fe_2_O_3_ rather than magnetite (Fe_3_O_4_). Therefore it was decided that all subsequent G1@Fe_3_O_4_ samples would be produced at 80 °C. The coprecipitation method can yield MNPs in a size range of 2–25 nm but with a broad size distribution (usually greater than ±25% from the mean) [33]. The microemulsion technique can narrow the size distribution to within ±5% from the mean, at the expense of a more challenging purification procedure and a much smaller quantity of product [33].

In our work, a slight but measurable improvement in size distribution from ±20% to ±16.5% from the mean was observed and the crystallites appeared separated by an organic layer (bright stripe around the dark cores—Figure 2c,d), which is consistent with their better dispersibility as compared to bare inorganic grains in close contact, that experience strong van der Waals attraction and usually cannot be separated. Aside from this moderate increase in crystallite size, it will be demonstrated that the main advantage of a synthesis templated by arborescent PS-*g*-P2VP copolymer micelles is the possibility of maintaining stability within a cell culture medium, which is the main challenge for using MNPs in biological applications.

The Bragg diffraction pattern obtained by selected area electron diffraction (SAED) confirmed a spinel structure corresponding to either magnetite Fe_3_O_4_ or maghemite γ-Fe_2_O_3_ (Figure 2e). Unfortunately, these two phases cannot be distinguished due to their isostructural character: both are cubic and have similar unit cell parameters. A few different peaks for these two iron oxides can appear but with a low intensity and easily blurred by background noise or crystallite size effects [34].

TGA was used to determine the relative amounts of iron oxide and polymer template in the nanocomposite particles. Decomposition of the G1 polymer started at ca. 350 °C and ended at ca. 450 °C (green dotted curve, Figure 2f). The thermogravimetric trace for a physically blended mixture of dry G1 powder and Fe_3_O_4_ paste (blue, long-dashed) was similar to that of the pure G1 sample. In comparison, the thermal decomposition (in N_2_) or combustion (in air) of the templated MNPs G1@Fe_3_O_4_ was characterized by a three-stage weight loss profile, more clearly visible in the differential weight loss curve for G1@Fe_3_O_4_ (Figure 2g) as follows:A small weight loss at 140 °C, attributed to the elimination of physically adsorbed water (bound to the surface of the MNPs) and surface hydroxyl groups [35,36], as also observed for the mixture of dry G1 powder and Fe_3_O_4_ paste (data not shown). This step differs from the evaporation of unbound water ending earlier, at ca. 130 °C. The slightly larger weight loss at 140 °C (solid trace, inset on Figure 2f) evidences the abundance of physically adsorbed water molecules and of surface hydroxyl groups, since the G1 template in G1@Fe_3_O_4_ was protonated.A massive weight loss in the 300–470 °C range, attributed to the decomposition of the PS and P2VP components in the G1 template not coordinated with Fe_3_O_4_. The rate of weight loss decreased before reaching 470 °C (Figure 2g). This weight loss is similar to what was observed for the two other samples, where the G1 template had no chemical interactions with Fe_3_O_4_.A weight loss that accelerated rapidly with the introduction of O_2_ at 470 °C (Figure 2g), suggesting burning of the P2VP phase coordinated with the Fe cations.

Similar stepwise thermal degradation of polymers in magnetic hybrids has been previously reported [35,37]. In our work, at 460 °C a portion of the ligands immobilized at the surface of MNPs decomposed, while more energy (a higher temperature) was needed to break the bonds formed between P2VP blocks and Fe_3_O_4_. The stepwise thermal degradation of the G1 template likewise evidences strong interactions between P2VP and Fe_3_O_4_, thus confirming the success of templating.

Beyond an initial 3.1 wt% loss due to the elimination of water (T ≤ 140 °C), a 54.9 wt% loss of organic material was observed for the G1 template for T ≤ 470 °C and a residual mass of ca. 42.0 wt% was found at the end of the TGA experiment. If the residue were composed of pure γ-Fe_2_O_3_, this would correspond to 43.3 wt% γ-Fe_2_O_3_ after correction for the initial water loss. This is higher than the theoretical composition of 35.1 wt% Fe_2_O_3_ calculated for a Fe/2VP molar ratio = 0.7. The γ-Fe_2_O_3_ content overestimation is attributed to incomplete degradation/combustion of the organic material in TGA analysis, leaving some carbonaceous material within the residue, as reported before in the literature [36].

The actual mass of γ-Fe_2_O_3_ in the residue was determined by UV-VIS analysis in 5 M HCl after dissolution of the solid into Fe^3+^ cations, complexed by Cl^−^ anions. The residue (ca. 1.73 mg) was not completely soluble in 5 M HCl, leaving some black particles clearly evidencing the presence of organic material. The equivalent Fe content was determined from a linear calibration curve (Appendix A) relating the iron concentration to the optical density at λ = 350 nm (OD_350_) as suggested by Arosio et al. [38]. The absorbance peak at this wavelength is ascribed to the hexachloride complex [FeCl_6_]^3−^ in 5 M HCl medium. An OD_350_ = 0.3695 was obtained for the sample, corresponding to 1.29 mg γ-Fe_2_O_3_. The total sample weight loaded for the TGA analysis was 4.126 mg, of which 3.1 wt% was water, leaving 3.998 mg of nanocomposite. The γ-Fe_2_O_3_ content based on UV-VIS analysis of the residue is therefore 1.29/3.998 = 32.2 wt%, reasonably close to the theoretical composition of 35.1 wt% Fe_2_O_3_. As compared to TGA, the UV-VIS analysis method was therefore considered more reliable to determine the equivalent Fe (and Fe_3_O_4_) content of the MNPs.

### 3.2. Stabilization of Magnetic Nanoparticles by the Polyion Complexation Technique

Double-hydrophilic block copolymers (DHBC) with a long PHEA segment, PAA_13_-*b*-PHEA_150_ and PAA_27_-*b*-PHEA_260_, proved to be the most efficient stabilizers among the DHBC series tested, as they could form stable unimolecular micelles with the G1 copolymer, even at low *f* = CO_2_H/N ratios [27]. These DHBCs were therefore tested to form micelles with G1@Fe_3_O_4_ by the same procedure applied to G1 alone. As expected, stable magnetic polyion complex (MPIC) micelles were formed for the same *f* ratios as when using the bare copolymers. PAA_13_-*b*-PHEA_150_ assembled with G1@Fe_3_O_4_ for *f* = 1 to form MPIC micelles with an intensity-weighted *D*_h_ = 124 nm (Figure 1b) and a number-weighted $D_{h}^{n}$ = 66 nm, PDI = 0.172. Increasing the DHBC amount further yielded MPIC micelles in the same size range, with PDI = 0.175. A reduced quantity of DHBC stabilizer at *f* = 0.5 almost led to doubling of the size of the MPIC micelles (*D*_h_ = 181 nm, $D_{h}^{n}$ = 118 nm, PDI = 0.150). Therefore *f* = 1 was considered the optimal ratio for the complexation of G1@Fe_3_O_4_ with PAA_13_-*b*-PHEA_150_. MPIC micelles for *f* = 1 and 0.5 were nevertheless both tested for biocompatibility.

PAA_27_-*b*-PHEA_260_ also assembled with G1@Fe_3_O_4_ to form stable MPIC micelles (Figure 1c,d). MPIC micelles with *D*_h_ = 126 nm, $D_{h}^{n}$ = 83 nm and PDI = 0.136 were obtained for *f* = 0.5, the optimal ratio observed for complexation with the G1 substrate. While a ratio *f* = 0.25 failed to prevent aggregation in complexation with G1 alone, it was able to stabilize G1@Fe_3_O_4_ and yielded MPIC micelles in the same size range ($D_{h}^{I}$ = 128 nm, $D_{h}^{n}$ = 62 nm and PDI 0.166) as for *f* = 0.5. The presence of Fe_3_O_4_ in the G1@Fe_3_O_4_ structure likely increased the volume of the G1 template, exposing more coordination sites and thus resulting in more efficient complexation even at the lower *f* ratio. The lower stability limit for PAA_27_-*b*-PHEA_260_ DHBC was determined to be *f* = 0.25, since the MPIC micelles obtained for *f* = 0.125 aggregated further (*D*_h_ = 235 nm and PDI = 0.322). The samples obtained for *f* = 0.5 and 0.25 were tested for biocompatibility.

MPIC micelles at *f* = 0.5 appeared as aggregates of Fe_3_O_4_ NPs on the TEM images, with *D*_TEM_ = 11.6 ± 2.0 nm for the Fe_3_O_4_ crystallites (Appendix A). When observed by AFM, the MPIC micelles appeared as spheres with a rather broad size distribution on the mica surface. A soft hydrophilic shell surrounding each MPIC micelle was visualized around a harder core on the phase image (Figure 3b), confirming the presence of a PHEA layer. A larger phase lag in tapping mode AFM is indeed associated with a softer material [39,40]. The inner core–shell morphology of the MPIC micelles was probed using different amplitude set-points (Appendix A), that is, different forces exerted by the cantilever on the surface.

The zeta potential of the MPIC micelles G1@Fe_3_O_4_@PAA_27_-*b*-PHEA_260_ at *f* = 0.5 was found to be pH-dependent, similarly to the metal-free PIC complexes discussed in our previous work [27]. Highly negative charges (ca. −20 mV) were found at pH ≥ 7 (Figure 3c) due to the presence of CO_2_^−^ functionalities. In the highly basic range, a plateau in the zeta potential of the MPIC micelles indicates complete ionization of the CO_2_H groups. The increase in zeta potential in the acidic pH range is ascribed to protonation of the carboxylate groups to produce more hydrophobic CO_2_H groups, while the 2VP units of the G1 substrate yield NH^+^ moieties. The isoelectric point (IEP) of the MPIC micelles was around pH 3. 

### 3.3. pH Responsiveness of MPIC Micelles

At pH 7.4 the intensity-weighted hydrodynamic diameter was 130 nm, almost identical with *D*_h_ = 126 nm at pH 7, the small change being possibly due to slightly higher ionization of the CO_2_H groups leading to stronger electrostatic repulsion among adjacent CO_2_^−^ groups. Slight shrinking in the high pH range (8–9.5) is ascribed to the increased ionic strength. Adjusting the pH to acidic also reduced the hydrodynamic size of the MPIC micelles, this time due to protonation of the CO_2_^−^ groups to produce CO_2_H. Neither clustering nor destabilization was observed over the whole pH range, indicating good colloidal stability of the MPIC micelles. At the isoelectric point (IEP), at pH ~ 3, the MPIC micelles were also stable with *D*_h_ = 104 nm. Disassembly of the MPIC micelles into their individual components (polymers and MNPs) was not observed at any pH tested. The changes in zeta potential and size were reversible over the whole pH range. The presence of Fe_3_O_4_ in the MPIC micelles apparently improved the colloidal stability of the hybrid MNPs as compared to the PIC micelles obtained at the same *f* ratio. This suggests that the PAA block of the DHBC forms a complex not only with the P2VP segments but also with the Fe_3_O_4_ phase, in agreement with previous work showing that PAA chains adsorb strongly onto iron oxide MNPs, endowing them with very efficient steric stabilization in cell culture media [41].

### 3.4. Biocompatibility

#### 3.4.1. Cytotoxicity

The cytotoxicity of the MPIC micelles – if significant – could be due to their hybrid polymeric and iron oxide composition. The cytotoxicity of bare Fe_3_O_4_ MNPs has been reported to be dose-dependent: at 2000 µg of Fe_3_O_4_ SPION/mL cell viability dropped to 40% as compared with the control [16]. In our work, the bare G1@Fe_3_O_4_ particles were also found to have dose-dependent cytotoxicity. At the highest concentration tested (1250 µg Fe_3_O_4_ /mL) for a long exposure time of 48 h, L929 cell viability was 51 ± 8% (Figure 4a non-coated, *f* = 0). The relative viability was higher (57 ± 3% and 81 ± 4%) when treated with 700 and 140 µg of Fe_3_O_4_/mL suspensions, respectively. These results are analogous to the toxicity of bare Fe_3_O_4_ MNPs reported in the literature, indicating that the G1 template had no significant effect on L929 cell viability in itself despite its polycationic nature.

Fe_3_O_4_ MNPs coated with flexible and hydrophilic polymers to prevent opsonisation, thus extending the circulation time of the MNPs, were found to be relatively nontoxic (even at high concentrations) but this result strongly depended on the type of cell tested and the chemical nature of the coating. Thus 99 to 92% viability was reported for telomerase-immortalized primary human fibroblast cells relatively to the control after exposure to PEG-coated MNPs (1 mg Fe_3_O_4_/mL, 24 h) [16] and to pullulan-coated Fe_3_O_4_ MNPs (2 mg of Fe_3_O_4_/mL, 24 h) [42], respectively. After 24 h of incubation with 170 µg of Fe_3_O_4_/mL suspension of amino-SPIONs, less than 40% of human melanoma metastasis Me237 cells were viable; the viability reported for Me275 cells was below 20% [43]. For 550 µg of Fe_3_O_4_/mL, MPEG-Asp_3_-NH_2_-coated iron oxide MNPs had no significant cytotoxicity for OCTY mouse cells, while MPEG-PAA-coated and PAA-coated iron oxide MNPs significantly reduced cell viability, resulting in 84% cell loss [44]. Regarding in vivo studies, it was reported by Cole et al. that the PEGylation of starch-coated MNPs led to decreased accumulation in the liver and kidneys in rats and enabled their crossing of the blood-brain barrier to accumulate in a glioblastoma brain tumour under magnetic attraction [45]. Such a spectacular escape of the reticuloendothelial system (RES) was also reported in rats for iron oxide MNPs at a high grafting density of dendronized, that is, arborescent, PEG chains [46].

The cytotoxicity of four types of MPIC micelles was investigated in the current work. Two of them were the MNPs G1@Fe_3_O_4_ coated with the PAA_13_-*b*-PHEA_150_ block copolymer at two different ratios, *f* = 1 (optimal in term of colloidal stability) and *f* = 0.5, while the other two were for the same magnetic core complexed with PAA_27_-*b*-PHEA_260_, having larger block sizes, at *f* = 0.5 (optimal) and *f* = 0.25. Low cytotoxicity was observed for the MPIC micelles up to the highest concentration tested (1250 µg Fe_3_O_4_/mL) after 48 h of incubation. Specifically, over 91 ± 7% cell viability was observed for the MPIC micelles as compared with 51 ± 8% cell viability for the bare MNPs. Even after a long incubation period (48 h), the L929 cells were well-spread in the wells and some were still undergoing mitosis, indicating their ongoing capability to proliferate after exposure to the MNPs. The cell loss ratios were negligible at reduced Fe_3_O_4_ concentrations (700, 140, 70, 28 and 14 µg of Fe_3_O_4_/mL). The MPIC micelles formed at optimal coating ratios caused no or negligible reductions in cell viability. Coating of the MPIC micelles with a less dense polymeric shell led to slightly higher reductions in cell viability but only at the higher concentrations tested. These results show that PAA-*b*-PHEA successfully formed a protective, hydrophilic and biocompatible shell around the Fe_3_O_4_ NPs.

#### 3.4.2. Cell Internalization Studies

Preliminary qualitative cellular uptake studies were done with the L929 cell line using confocal laser scanning microscopy. To this end cytoskeleton *F*-actin was labelled with phalloidin (red), while the nuclei were labelled with DAPI (blue). The MPIC micelles labelled with the fluorescent probe fluoresceinamine (MPIC* micelles) were visualized as green fluorescent species. The cellular uptake of the current MPIC* micelles appeared identical with previously studied PIC micelles (*D*_h_ = 42 nm, PDI = 0.08) [27] even though their diameter was larger (*D*_h_ = 104 nm, PDI = 0.14). While absent in images of the untreated cells (Appendix A), the MPIC* micelles were mainly located at the periphery of the nuclear membrane and probably in cytoplasmic vesicles (Figure 4d–h). The 3D reconstruction from a stack of 50 cross-sections in Figure 4h (also provided as Appendix A) clearly shows that the MPIC* micelles are located inside the cytoplasm, although some also stayed at the periphery of the nuclear membrane. The appearance of spherical green fluorescence dots below optical resolution inside those compartments supports the stability of MPIC* micelles in a low pH environment, for example, in lysosomes and endosomes, in the presence of degrading enzymes. Significant clustering after the endocytosis of iron oxide MNPs in the intracellular environment has been suggested as a reason for the enhancement of magnetic properties and increased magnetic resonance imaging contrast [47]. TEM was also used to confirm the internalization of MPIC micelles and to determine the distribution of MNPs after cellular uptake. Iron oxide crystallites with high electron contrast appeared inside large intracellular endosomes or lysosomes, visible in a clearer area that could be differentiated from the grey background of the cytoplasm (Figure 4i and Appendix A). This location of the iron oxide crystallites is in good agreement with the results reported in the literature [47,48], suggesting that MPIC micelles could be also internalized through the cell membrane by clathrin-dependent endocytosis (Appendix A), although the formal proof of this internalization pathway would require complementary experiments with clathrin inhibitors as in Sanchez et al. [49].

Cell internalization was also quantified by the fluorescence-activated cell sorting (FACS) assay. The number of fluorescence-positive cells counted per 10,000 cells, expressed as % positive cells and the ratio of fluorescence intensity of 10,000 treated cells to that of 10,000 untreated cells, expressed as their mean fluorescence intensity (MFI), were extracted from the FACS data to evaluate the extent of cellular uptake. The extent of cellular uptake exhibited a strong dependence on MPIC micelle concentration (Figure 4j,k). Even at low MPIC micelle concentrations (1.4 µg Fe_3_O_4_/mL), 0.6% of the cells were reported as fluorescence-positive, emitting a low MFI signal (MFI = 69). The percentage of positive cells and their MFIs increased for increasing MNP incubation concentrations. Exposure to a higher concentration (70 µg Fe_3_O_4_/mL) was sufficient for 80.5% of cells to uptake the fluorescently labelled MPIC* micelles, emitting 624 MFI units. A smaller increase in % positive cells (from 80.5% to 98%), as compared with a ca. 40% increase of the MFI signal (from 624 to 1057) was observed when doubling the Fe_3_O_4_ incubation concentration from 70 to 140 µg·mL^−1^, indicating that positive cells continued to uptake more MNPs, which led to variable amounts of MNPs among the cells. This is in good agreement with the uneven internalization observed by confocal imaging. A similar phenomenon was reported by Asín et al., who observed a slight change in the % positive cells accompanied by a large increase in the MFI values when incubating cells at a higher MNP concentration [50].

The incubation time was an important parameter that also influenced cell internalization. Detectable MFI signals from the cells only 1 h after treatment with 140 µg of Fe_3_O_4_/mL solution demonstrated the rapid uptake of MPIC micelles by L929 cells. Extending the incubation period further enhanced the cellular uptake. After 12 h of incubation, 45.6% of the cells fluoresced with a MFI = 444 units while after 24 h, these values were 97.2% and 1067 MFI units. Linear regression of MFI and % positive cells versus incubation time data suggests a steady rate of MNP internalization within the time interval tested, which could be used as a predictive tool for cellular uptake kinetics (Figure 4n,o).

### 3.5. In Vitro Cellular Radiofrequency Magnetic Field Hyperthermia

As compared to normal cells which can withstand temperature rises (i.e., up to 42 °C) during hyperthermia treatment, tumour cells were discovered to have a disorganized and abnormal vascular environment, resulting in poor heat dissipation capability and are thus more sensitive to heat-induced-apoptosis [51]. In almost all in vivo studies of hyperthermia treatments using MNPs as heat mediators, a suspension of MNPs is injected directly into the tumour, typically at a concentration of several tens of mg·mL^−1^ [52], allowing the MNPs to be internalized by the cancer cells or at least to be in the tumour microenvironment (extracellularly) [53]. The heat produced by the oscillating magnetic moments of the MNPs exposed to an external alternating magnetic field (AMF), usually at a frequency of several hundreds of kHz, lead to cell death. It is important to categorize published in vitro magnetic hyperthermia research results into magnetic fluid hyperthermia, where cells are dispersed in a magnetic fluid during AMF exposure and intracellular magnetic hyperthermia, where heat is exclusively generated by internalized MNPs. The lower efficiency of heating induced by intracellular MNPs as compared to freely dispersed MNPs in water is likely due to confinement of the MNPs within the intracellular vesicles, implying that Néel relaxation is the sole mechanism for heat generation by intracellular MNPs [30].

It is also believed that cancer cells can only be killed when the temperature rises over 43 °C, defining the cumulative effective thermal dose at 43°C or CEM43. However, recent work by Creixell et al. [54] and Asín et al. [50] demonstrated the possibility of inducing significant cell death with internalized MNPs without observing a macroscopic temperature increase during AMF exposure, a puzzling effect sometimes called “intra-lysosomal” or “cold hyperthermia.” The absence of macroscopic heating by MNPs localized within cells suspended in an aqueous buffer with good heat conductivity also makes sense from a physics point of view, when comparing the potential heating power relatively to the much larger heat losses on a typical cell scale of 10 µm, [55]. Furthermore, the possibility of causing a detectable temperature rise at a cellular level is still being debated within the cell imaging community [56]. In a study by Creixell et al. [54], only 50% of the cells incubated with Fe_3_O_4_ MNPs were viable after AMF exposure (233 kHz, *H* = 37.5 kA·m^−1^, 2 h). In comparison, EGFR-targeting Fe_3_O_4_ MNPs resulted in 40% of cell viability, even in the absence of a magnetic field. Exposing the cells incubated with EGFR-targeting Fe_3_O_4_ MNPs to the same AMF treatment reduced cell viability to 4–6%, evidencing the synergistic effect of specific targeting receptors combining antibodies and magnetic hyperthermia. Asín et al. also reported 100% cell death by exposing cells incubated with MNP-loaded dendritic cells to an AMF (260 kHz, *H* = 12.7 kA·m^−1^, 15 min) [50]. Another targeted cellular hyperthermia experiment was reported by Sanchez et al.: when gastrin, a peptide sequence targeting hormone-dependent endocrine pancreatic cancer receptor CCK2, was grafted onto iron oxide MNPs, their internalization in lysosomes reduced cell survival to 30% under AMF exposure (275 kHz, *B* = 52 mT, *H* = 41.6 kA·m^−1^, 2 h) [49].

Since our experiments showed that MPIC micelles could be easily internalized by L929 cells, we studied their use in intracellular magnetic hyperthermia on both L929 and U87 cells. Because the amount of internalized MPIC micelles was strongly dependent on the incubation concentration, the cells were incubated with MPIC micelle solutions at Fe_3_O_4_ concentrations of 1250, 700 and 140 µg·mL^−1^ in complete DMEM culture medium for 15 h. The cells were then extensively rinsed with PBS to ensure that all the MPIC micelles not taken up (e.g., simply adsorbed on cell membranes) were removed and that the hyperthermia effect, if present, was only due to intracellular MNPs. The cells were suspended in culture medium during AMF exposure (755 kHz at *H* = 10.2 kA·m^−1^, for 1.5 or 3 h) before they were seeded for the cell viability assays. The control cells were incubated in complete medium and exposed to the same AMF.

The temperatures of the control sample and the incubated sample were monitored during the whole AMF exposure and, in both cases, no increase in temperature was observed (Figure 5a), indicating that there was no parasitic heating due to eddy currents in the high ionic strength buffer and that the amount of heat dissipated by the intracellular MNPs was also insufficient to increase the temperature of the whole sample. This observation seems to confirm our hypothesis that AMF excitation of the intracellular MNPs was the main factor leading to cell death, without macroscopic heating of the bulk medium.

Cell viability had a strong dependence on both the incubation concentration and the exposure time (Figure 5b,c). For a concentration of 140 µg Fe_3_O_4_/mL, ca. 71–76% of the cells of both types survived after 3 h of AMF exposure. Increasing the incubation concentration to 700 µg/mL reduced cell viability to 72–77% after 1.5 h and to 38–54% after 3 h of magnetic field application. At 1250 µg/mL (the highest concentration at which cell cytotoxicity in the absence of field can be neglected), lower cell viabilities were observed: about 46–57% of cells remained after 1.5 h of treatment and 30–35% remained after 3 h. The drop in cell viability at higher incubation concentrations is ascribed to the larger number of MNPs internalized in the cells.

Apoptosis and lysosomal membrane permeabilization due to the production of reactive oxygen species (ROS) [49,50,57,58,59] have been suggested as causes leading to cell death. The rationale is that it is the release of ROS from the lysosomes, made permeable by the combination of MNP uptake and AMF exposure, that kills the cells rather than a temperature increase (insignificant at the cellular level). More precisely, Asín et al. [50] proved that AMF-treated dendritic cells released a toxic agent to the neighbouring cells (not exposed to the magnetic field), while Connord et al. [57] built a setup to image the cells under AMF treatment by CLSM and showed by fluorescence colocalization that the lysosomes progressively released their gastrin-grafted MNPs during the application of AMF (300 kHz, *B* = 53 mT, *H* = 42.4 kA·m^−1^, 1.5 h). In our study, lysosomal membrane permeabilization due to local heating is deemed to be the most likely mechanism leading to cell death because (i) macroscopic heating was absent; and (ii) MNPs were found within vesicles, probably late endosomes or lysosomes, after they were internalized through the cell membranes. Extending the AMF exposure time resulted in killing more cells, probably due to the variable number of MNPs internalized observed by FACS. The cells internalizing a sufficiently large number of MNPs should be more sensitive to AMF exposure, while cells with fewer MNPs may be killed only after longer AMF exposure times.

## 4. Conclusions

For the first time, unimolecular micelles made from a G1 arborescent copolymer (G0PS-*g*-P2VP) were used as templates to control the size and improve the size distribution of iron oxide MNPs. The templating effect of G1 was evidenced by (i) the formation of larger Fe_3_O_4_ crystallites with *D*_TEM_ ≈ 12 ± 2 nm, (ii) enhanced colloidal stability brought by electrostatic repulsion from protonated NH^+^ groups of the P2VP chains, (iii) a slight improvement in the size distribution as compared to non-templated MNPs and (iv) the separation of crystallites appearing in the TEM images by an organic layer. This structure has many positive effects on the properties, such as yielding a higher specific absorption rate (SAR) in magnetic hyperthermia and a higher relaxivity, that is, a higher magnetic resonance imaging contrast (these two features being the subject of a forthcoming paper) and better dispersibility (inorganic grains in close contact usually experience strong van der Waals attraction and cannot be separated, whereas in MPIC micelles they are coated by an organic shell). The three-stage weight loss thermal decomposition and combustion profile observed for the templated MNPs G1@Fe_3_O_4_ was comparable to previously published results, evidencing strong interactions between P2VP and Fe_3_O_4_ and thus successful templating. UV-VIS analysis in 5 M HCl was more reliable than TGA in determining the equivalent Fe (and corresponding Fe_3_O_4_) content of the MNPs.

The MPIC micelles obtained by complexation had a core–shell morphology, with a hydrodynamic diameter *D*_h_ ≈ 130 nm (PDI ≈ 0.136) and good colloidal stability, not only under physiological conditions (at pH 7 with salts) but also over a wide pH range, which shows great promise in terms of biocompatibility when biomedical applications are being targeted. The MPIC micelles were non-cytotoxic up to the highest concentration tested (1250 µg Fe_3_O_4_/mL) after 48 h of incubation. MPIC* micelles were successfully internalized in L929 cells, while cellular uptake exhibited strong dependence on the NP concentration and incubation time.

An in vitro MFH assessment conducted on L929 cells widely used for cytotoxic assays of nanomaterials and human brain cancer U87 cells representing the main medical target of magnetic hyperthermia for humans [29], using custom-built MFH equipment and a newly proposed cell treatment protocol, revealed a large decrease in cell viability as a function of both the incubation concentration and the exposure time to the AMF, evidencing a dual dose-effect. At the maximum incubation dose (1250 µg/mL iron oxide), the lowest cell viabilities 30–35% were observed after 3 h of AMF exposure. The hypothesis that AMF excitation of the intracellular MNPs was the main factor leading to cell death was verified, based on the fact that cell viability decreased in a dose-dependent manner with the concentration of internalized MNPs in the cells, even in the absence of macroscopic heating, as expected for intracellular magnetic hyperthermia. While the two cell lines are very different, they started responding at a low incubation concentration of 140 µg/mL Fe_3_O_4_ (100 µg/mL Fe), three orders of magnitude lower than the dose of 160 mg/mL Fe_3_O_4_ (112 mg/mL Fe) used for intratumoral instillation in clinical assays reported on human brain tumours [29].

## Figures and Tables

**Figure 1 nanomaterials-08-01014-f001:**
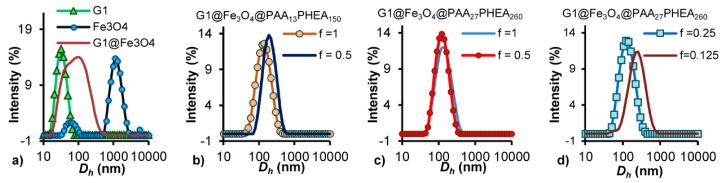
Hydrodynamic size distribution for (**a**) G1 (pH 4), Fe_3_O_4_ (pH 1.4) and G1@Fe_3_O_4_ (pH 1.4). (**b**–**d**) MPIC micelles G1@Fe_3_O_4_@PAA-*b*-PHEA (pH 7) for different *f* ratios at 25 °C.

**Figure 2 nanomaterials-08-01014-f002:**
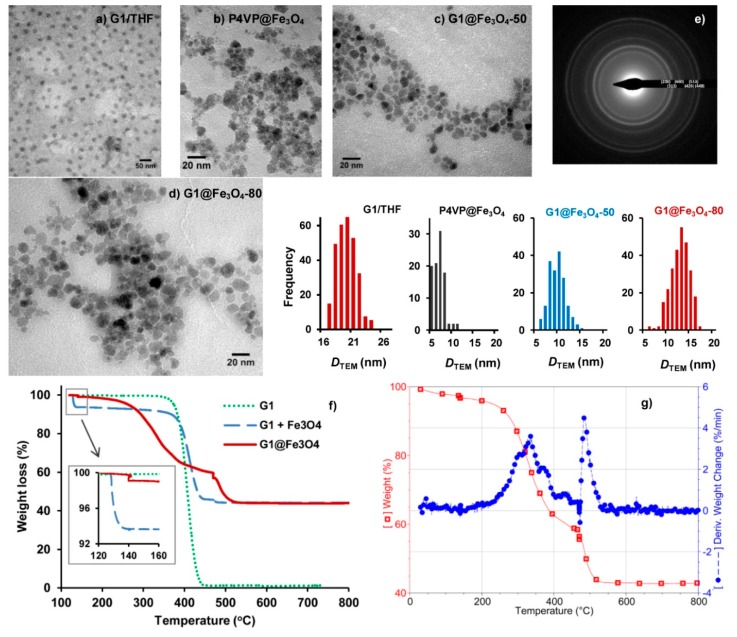
(**a**) TEM image of G1 arborescent PS-*g*-P2VP micelles. Crystallites of Fe_3_O_4_ NPs synthesized in the presence of (**b**) linear P4VP chains at 50 °C, (**c**) G1 at 50 °C and (**d**) G1 at 80 °C. (**e**) SAED for G1@Fe_3_O_4_ with assignment of diffraction rings to the Bragg peaks for Fe_3_O_4_. (**f**) TGA curves for pure G1 template (green, dotted), a mixture of dry G1 and Fe_3_O_4_ (blue, long dash) and G1@Fe_3_O_4_ (red, solid). (**g**) Weight loss (red, 
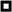
) and differential of the weight loss curve (blue, ●) for G1@Fe_3_O_4_.

**Figure 3 nanomaterials-08-01014-f003:**
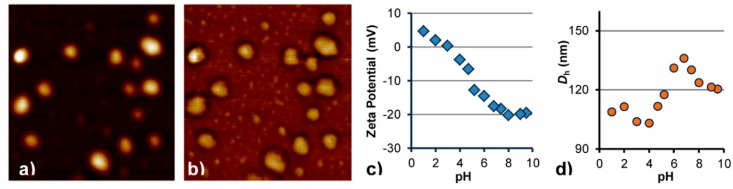
AFM (**a**) height and (**b**) phase images for MPIC micelles G1@Fe_3_O_4_@PAA_27_-*b*-PHEA_260_ at *f* = 0.5. The scale of the AFM images is 500 × 500 nm^2^. (**c**) Zeta potential and (**d**) Intensity-weighted hydrodynamic diameter *D*_h_ of mentioned MPIC micelles as a function of pH.

**Figure 4 nanomaterials-08-01014-f004:**
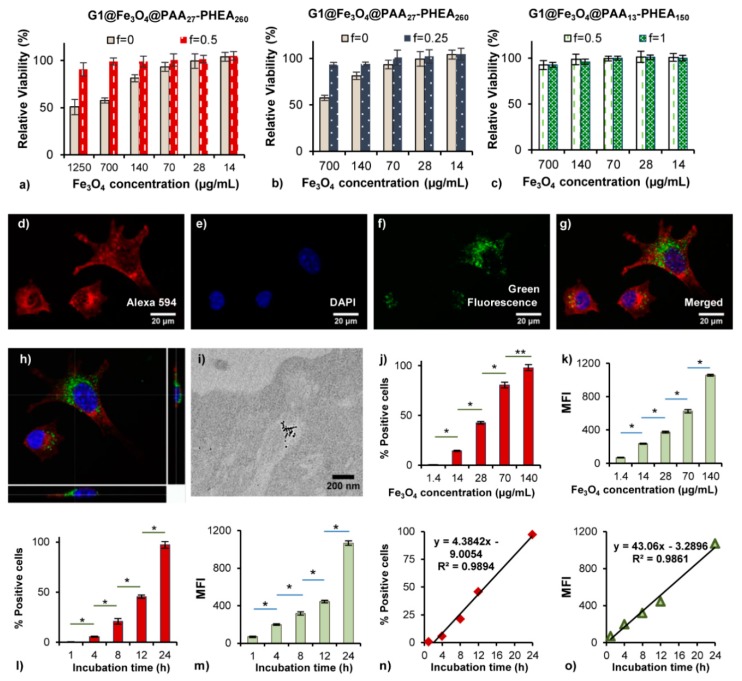
(**a**–**c**) Cytotoxicity profiles for uncoated MNPs G1@Fe_3_O_4_ (*f* = 0) and MPIC micelles G1@Fe_3_O_4_@PAA-*b*-PHEA at various complexing ratios *f* (L929 cells, 48 h of incubation). Cell viability is expressed as the mean value and standard deviation for three independent experiments with four replicates, relatively to the untreated cells (100% control). (**d**–**h**) Confocal microscopy images for L929 cells treated with the green fluorescent MPIC* micelles G1@Fe_3_O_4_@PAA*_27_-*b*-PHEA_260_
*f* = 0.5; 140 µg Fe_3_O_4_/mL, 24 h incubation. (**i**) TEM images for L929 cells treated with an MPIC micelle solution (G1@Fe_3_O_4_@PAA_27_-*b*-PHEA_260_
*f* = 0.5; 100 µg/mL, 24 h incubation). (**j**–**m**) Dependence on the concentration of the mean fluorescence intensity (MFI) for L929 cells treated with MPIC* micelle G1@Fe_3_O_4_@PAA*_27_-*b*-PHEA_260_
*f* = 0.5 solutions (after 24 h incubation) and on the incubation time (140 μg Fe_3_O_4_/mL concentration; * *p* < 0.002, ** 0.002 < *p* < 0.01 by the two tails Student’s *t*-test, degrees of freedom = 4). (**n**,**o**) Linear regressions for the MFI and % positive L929 cells vs. incubation time.

**Figure 5 nanomaterials-08-01014-f005:**
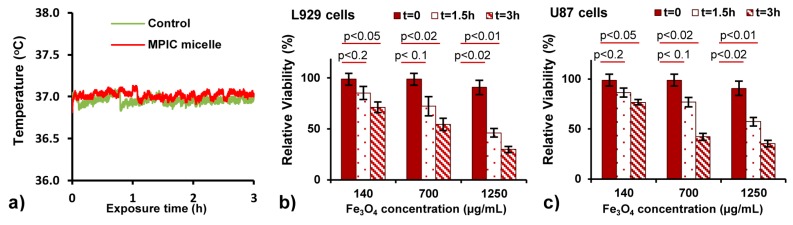
(**a**) Temperature variation in suspensions of control cells and cells incubated with MPIC micelles during exposure to an alternating magnetic field (755 kHz, 10.2 kA·m^−1^). Viability of L929 (**b**) and U87 (**c**) cells determined by the MTS assay after 24 h of incubation with MPIC micelles G1@Fe_3_O_4_@PAA_27_-*b*-PHEA_260_
*f* = 0.5, with or without exposure to the high frequency alternating magnetic field. Cell viability is expressed as the mean value relatively to untreated cells (control 100%; probability p was calculated by the two tail Student’s *t*-test).

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
