# Peer review of "Magnetic Polyion Complex Micelles for Cell Toxicity Induced by Radiofrequency Magnetic Field Hyperthermia"

_nanomaterials, 2018, doi:10.3390/nano8121014_

Reviewer 1 Report

The manuscript entitled "Magnetic Polyion Complex Micelles for Cell Toxicity Induced by Radiofrequency Magnetic Field Hyperthermia" authored by Vo Thu An Nguyen et al. reports on synthesis and characterisation of magnetic polycomplex (MPIC) micelles  for Hyperthermia Study. The paper demonstrates that MPIC micells are colloidally stable in acqueous environment over different pH range values. They are uptaken by cells and they show a dual dose-dependent (concentration and duration of radiofrequency magnetic field exposure) effect on cytoviability.The paper is well structured and written in a good English. Accordingly, it requires only minor changes before beeing acceptable for publication in Nanomaterials since it is  also rather interesting and relatively novel for its readers. Suggestions\changes\improvements are listed below:

1) A slight English revision to improve language and to correct few typos\imprefections is reccomended.

2) The manuscript text length can be slightly reduced for better comprehension for the readers.

3) A references literature update  is also suggested (eg by adding few  recent reviews of the group of T. Pellegrino and\or articles in the area of  Iron Oxide NPs as drug delivery means such as Mancarella et al. Macrom. Biosc. 2015)

4) What was due the choice of cancer cells? Are healthy cells also taken in consideration as control? Please comment\discuss on it.

5) What about possible in-vivo studies? Did authors plan them? Outlook and perspectives at the end of the manuscript are suggested. Authors please add\update discussion on these issues

6) How was done MPIC internalisation studies? Did the author have\plan to  quantified\quantify it through selected co-localisation parameters by Confocal Microscopy\FACS? Please add comment on this issue.

7) Did author plan to investigate drug loading\release originated by these MPIC against tumor cells? Discuss\Comment on it.

Reviewer 2 Report

Work is interesting, well thought in the part of experiments and it is reasonably illustrated. It has a potential to be in a future a very good reference work but some aspects of the manuscript require a major revision in order to put discussion at the level of the planned research. Below I propose some changes in order to improve manuscript up to the highest level.

Despite the fact that lis of the references is extensive some aspects are not really covered. For example, ferrogel encapsulation or natural polymer steric stabilization can be viewed as an alternative to the  MPIC micelles (Prog. Polym. Sci. 2006, 31, 603–632; Sensors 2017, 17(11), 2605; Nanoscale 2015, 7, 9686–9693, etc.) comparative overview might be quite attractive. 

Introduction contains no clear description of the goal of the present studies - it is necessary to re-write the final part of the introduction.

Manuscript contains no evidence that synthesized particles are magnetite - XRD data might be somehow substituted by TEM but without proper Red-Ox titration it is impossible to say what kind of iron oxide is under consideration (Nano Lett., 2009, 9 (10), pp 3651–3657).

particle size distributions must be mathematically fitted and parameters extracted and discussed properly. 

Additional comments are necessary with respect to "good" colloidal stability (zeta potential is only -20 mV) (AIP Adv. 2012, 2, 022154; Nanomaterials 2017, 7, 373; etc.).

Conclusions are very long and they contain part of discussion (with ref. etc.) - these two parts must be better separated. 

Reviewer 3 Report

The authors present an interesting study about a novel procedure for the synthesis of magnetic micelles sensitive for hyperthermia stimuli. The results are supported by several techniques and the conclusions, although not very concise, are acceptable.

Despite the quality of the research presented, the introduction needs to improve some points, especially in relation to the updating of bibliographic references. There are many reviews in this field from the last two years that should be included in the text.

Lines 38 to 48; 242 to 244 It is well known, as mentioned by the authors, that bare Fe3O4 nanoparticles are unstable and need to be coated with polymers to increase their stability. There are several groups that have obtained highly stable SPIONS with low size distribution that have not been considered in the text (see Estelrich et al.; De Cuyper M. et al.; Soenen, et al…).

Author Response

Round  2

Reviewer 2 Report

The manuscript was revised in accordance with my comments and it is now ready for publication.